# Dynamic of Vascular Streak Dieback Disease Incidence on Susceptible Cacao Treated with Composted Plant Residues and *Trichoderma asperellum* in Field

**Ade Rosmana** [1,*], **Muhammad Taufik** [2], **Asman Asman** [1], **Nurul Jihad Jayanti** [1] and **Andi Akbar Hakkar** [3]

1   Cocoa Research Group, Faculty of Agriculture, Hasanuddin University, Makassar 90245, Indonesia; asman_adi81@yahoo.com (A.A.); jayantinuruljihad@gmail.com (N.J.J.)
2   Plant Protection Department, Faculty of Agriculture, Halu Oleo University, Kendari 93232, Indonesia; taufik24@yahoo.com
3   Palu Agricultural Quarantine Center, Ministry of Agriculture, Palu 94231, Indonesia; andiakbarhakkar@yahoo.com
*   Correspondence: aderosmana65@gmail.com; Tel.: +62-816-279-147

**Abstract:** *Trichoderma asperellum*, composted plant residues, and its combination can control vascular-streak dieback (VSD) disease caused by fungus *Ceratobasidium theobromae* in laboratory conditions. In this trial, we evaluated these treatments in two years through the application of foliar spraying and stem infusion for *T. asperellum* alone, through soil amendment for compost alone, and *T. asperellum* plus this organic fertilizer in the field on susceptible cacao clone. The disease is characterized by full-leaf chlorosis and necrosis that can develop rapidly to the entire branch, with around 70% incidence in seven months, and we detected the pathogen in branches showing light symptoms. All treatments except for *T. asperellum* plus composted plant residues three and seven months post application did not have any impact on the reduction of VSD incidence in the first year. In the second year, we observed a significant reduction of the disease by both *T. asperellum* in combination with compost and compost alone in a time span of three to seven months, and with *T. asperellum* spraying and infusion in a time span of five to seven months. By comparing to the control, the efficacy of these treatments in controlling the VSD disease seven months post-first application in the last year was 44.4%, 23.5%, 23.1%, and 15.1%, respectively. Detection of trees inoculated with *T. asperellum* indicated that this fungus was present in root and branch tissues except for treatment through infusions, while in the uninoculated trees, *Trichoderma* was not present or was present at a very low level. These data showed that combination of *T. asperellum* and composted plant residues applied through soil amendment was able to control VSD disease effectively and could potentially be used at large scale to control this disease and other diseases infesting aerial parts of cacao, and to improve soil fertility.

**Keywords:** *Trichoderma*; disease; compost; soil amendment; foliar spraying; infusion

## 1. Introduction

Cacao (*Theobromae cacao* L.) suffers from many devastating diseases, causing a steady decline inproduction and a reduction in bean quality. One of the most prominent diseases is vascular streak dieback (VSD), caused by the basidiomycetous fungus *Ceratobasidium theobromae* (syn. *Oncobasidium theobromae*), mainly in the Southeast Asian and Melanesia Region [1,2]. Since its discovery in 1983 in

East Kalimantan, Indonesia, this disease has been distributed in almost all provinces of the country, to a similar extent as *Phytophthora* pod rot disease [3]. In the field, basidiospores of the pathogen produced under moist conditions disperse by wind and infect flush leaves at the branch and then colonize the xylem [1,4,5]. Through this vascular tissue, hyphae spread to adjacent leaves, causing leaf drop and killing the branches of susceptible cacao genotypes [4,6]. As a typical symptom, chlorosis with scattered islets of green tissue develops on a single leaf, usually on the second or third flush behind theshoot apex, three to five months after infection. In some cacao genotypes, necrotic lesions on margins and tips of infected leaves also develop as an atypical leaf symptom [1].

In the past seven years, in cacao trees infested by the VSD pathogen, endophytic fungi such as *Fusarium* and *Lasiodiplodia* are much detected. Four species of *Fusarium* including *F. decemcellulare*, *F. solani*, *F. equiseti*, *F. incarnatum*, and three species *Lasiodiplodia* consisting of *L. theobromae*, *L. pseudotheobromae*, and *L. hormozganensis* have been identified from branches, petioles, and leaves of cacao [7,8] This high infestation is apparently because the fungi, known as weak secondary pathogens [9,10], receive a benefit for their development by increased plant weakening as a result of intensive infection by VSD. Their presence offers two possibilities. First, they could compete with the pathogen to decrease VSD incidence, or second they could have an impact on disease severity. The two fungi genera have also been known to cause diseases on cacao such as dieback [8,11–13]. New research has indicated that infection by *Lasiodiplodia* provides symptoms resembling infection by the VSD pathogen, with chlorosis and necrosis on leaves [14].

The control of this VSD complex in the field demands a long-term plan because the pathogen's ability to colonize and survive in cacao tissues makes it difficult to remove from a given tree and field. Cultural practices of pruning diseased branches about 30–40 cm below the end of visible streaking symptoms [15,16] and application of triazole systemic fungicides can delay the disease progression [17], but the high cost and labor intensiveness of these practices makes them not economically efficient. We have investigated using the endophytic *Trichoderma asperellum* as a promising approach for reducing the impact of VSD disease [3,18]. This *Trichoderma* can deploy systemically to almost all parts of the cacao plant after application through foliar spraying, infusion, and soil drenching [19], and its control capability against VSD is improved when applied in combination with composted plant residues through soil amendment [18]. *Trichoderma* sets up symbiotic interactions with plants and, like other root-colonizing microorganisms such as rhizobia and mycorrhizae, these interactions can induce systemic resistance to plant diseases [20]. With systemic distribution in cacao tissues, *T. asperellum* could offer both indirect and direct modes of action against the VSD pathogen.

The present study describes a further application of *T. asperellum* on a susceptible cacao clone that was heavy infected by VSD in the field to evaluate if this fungus could remediate the diseased tree. We applied *T. asperellum* through soil amendment combined with plant-residue-based-compost and compared this with *T. asperellum* alone applied through stem infusion as well as foliar spraying and compost alone. We then observed the dynamics of VSD incidence in two successive years and also observed the ability of this *Trichoderma* to colonize branches and roots. Our working hypothesis is that the combination should show a higher reduction in VSD incidence than the *Trichoderma* and the compost alone. Such differentiation of disease reduction level should help to understand the relationship between each treatment and their suppression mechanisms.

## 2. Materials and Methods

### 2.1. Field Establishment

The trial was established in March 2017 at a cacao field severely infested by vascular dieback disease (VSD) in Patalassang Village, Bantaeng Regency, South Sulawesi. The only cacao clone planted in this farm was the Jalani clone. Activities including the observation of disease incidence and sampling of leaves, branches, or roots for identification of the pathogen, characterization of symptoms, proof

of *T. asperellum* deployment, and identification of other co-occurring fungi were carried out until December 2018.

### 2.2. Determination of Vascular Streak Dieback Pathogen and Symptoms

The Jalani cacao clone is only found in Bantaeng and Bulukumba regencies and has never been used as the subject of research. To prove that this clone was infected by the VSD pathogen, *Ceratobasidium theobromae* was isolated and determined, and its symptoms were also characterized. To isolate this pathogen, fragments of twig showing light symptoms were surface sterilized, and their bark was removed from the surface. Then, these fragments were cut into 0.5-cm sections and placed onto water agar (WA) in Petri dishes, and the growing mycelium was morphologically characterized. The symptoms were determined by the presence of dieback, chlorotic leaves, dark vascular discoloration in wood and leaf scars, and three blackened vascular traces on the surface of the leaf abscission scars.

### 2.3. Production of Trichoderma Inoculum for Field Application

To produce sufficient inocula for field application, *Trichoderma asperellum* strain ART-4/G.J.S. 09-1559 was grown on rice grain medium in a sterilized plastic bag. The medium was inoculated with five plugs of *T. asperellum* spores and mycelia derived from 5-day culture in PDA medium, and the bag was sealed, punctured with a fine sterile needle for aeration, and then incubated at a temperature of 25–27 °C and humidity of 80–90%. After five days, the culture was harvested, dried at a temperature of around 40 °C for two hours, and then air-dried for several hours. The product was applied in the field after grinding to form a smooth powder.

### 2.4. Preparation of Medium for Field Application

In this study, *T. asperellum* was applied through foliar spraying, infusion, and soil amendment combined with plant residues. The foliar spray was done using a 15-L knapsack sprayer with a nozzle having four holes, and the infusion was applied using an infuser made from a 1.5-L plastic bottle with a small hose attached to a disposable syringe and needle connected to its cap. Soil amendment was done after composting plant residues consisting of gliricidia leaf, billygoat weed, and rice straw. The composting process was carried out for a period one month where raw materials were previously chopped using a crop machine and covered with a plastic sheet. Aeration was assured by reversing these plant residues every two to three days, and followed by a one month-curing period.

### 2.5. Field Assessment

The experiment in the field to assess the VSD incidence was arranged in a randomized block with treatments consisting of composted plant residues applied through soil amendment, *T. asperellum* and composted plant residues through soil amendment, *T. asperellum* through stem infusion, and *T. asperellum* through foliar spraying. Each treatment consisted of four trees and was repeated four times, providing a total, including the control, of 80 trees.

Soil amendment was conducted one time by placing composted plant residues in four holes around one meter from the main stem, mixed with *T. asperellum* before application. In this treatment, 10 kg of compost was applied per tree, and there was 4 g *T.asperellum* per kg compost (or 40 g/tree). For stem infusion, the bottle containing 4 g/L/tree of *T. asperellum* was hung and reversed, permitting the suspension run into stem via the needle of the syringe inserted in a reverse V-cut pocket of bark. The application was made three times in ten-day intervals. For foliar spraying, the spray was specially directed to young flush and conducted three times, also in ten-day intervals, with the concentration of 4 g/L/tree. The same treatment and same frequency of application was repeated in the second year.

The impact of treatments was evaluated by observation of VSD incidence on branches, deployment of *T. asperellum* into roots and branches, and the occurrence of endophytic fungi in root and branch tissues. Four branches with two to three leaves showing symptoms and representing four wind directions were sampled for continuous and constant observation of VSD incidence. This incidence was

observed four times per year: one, three, five, and seven months post first application. The presence and colonization of *T.asperellum* as well as other fungi in branch and root tissues were observed in the same time, two months post first application.

### 2.6. Re-Isolation of T. asperellum and Identification Other Fungi in Branch and Root Tissues

Branch and root samples from trees treated two months post first application were cut into $1 \times 0.5$ cm$^2$ and 1 cm sections, respectively after removing their bark. Five sections of branch and root respectively were sterilized in 2.0% sodium hypochlorite for three minutes, 70% ethanol for two minutes, and vigorously washed several times in sterile distilled water before being placed onto PDA in Petri dishes. These Petri dishes were incubated at room temperature and examined every day for the presence of *Trichoderma* and other fungi. Colonization was calculated in per cent by determining the fungus present in five sections of branch or root. The fungi found in these sections were distinguished based upon the morphological characteristics of their cultures in PDA medium. In addition, with the aid of a light microscope, hyphae septation and branching as well as the size, shape, and color of conidiophores, phialides, and conidia were also observed. This assessment of characteristics was used in taxonomic keys for the identification of fungi [21,22].

### 2.7. Analysis

VSD incidence was calculated by using the formula *I = a/b × 100%* where *I* is incidence, *a* is the number of leaves showing VSD symptoms, and *b* is the total number of leaves observed on one branch. Then, the data were analyzed without any transformation, and the least significant difference was used for evaluating significant differences between the treatment means, while the colonization of *Trichoderma* and other fungi was not analyzed statistically.

## 3. Results

### 3.1. Pathogen and Symptoms of VSD

Isolation of fungus from cacao branches with leaves showing chlorotic and dieback symptoms indicated that we found the fungus growing slowly from the ends of branch sections that presented cottony, non-sporulating white mycelium on WA medium and had *Rhizoctonia*-like hyphae according to microscopic observation. The disease infested almost all branches of trees in the field with variation in incidence from light to severe, and the last can cause dieback. A typical symptom of VSD in the form of chlorosis with scattered islets of green tissue on leaves and black vascular streaking on wood was not found in the Jalani cacao clone. We only observed atypical symptoms in the form of full leaf chlorosis followed by necrosis and wood vascular tissues with more irregular and reduced number of black streaks. However, another typical symptom (i.e., three vascular discolorations on leaf scars and petioles) was seen very clearly (Figure 1). The characteristic fungus and symptoms mentioned above proved that VSD pathogen, *Ceratobasidium theobromae* infected the trees.

### 3.2. Impact of Trichoderma and Compost on VSD

Cacao trees of the Jalani clone in the field were around ten years old, and the VSD disease had infested almost all of the trees. The disease incidence on untreated trees increased relatively quickly, reaching 44.7% and 16.4% one month post first application and 68.6% and 69.2% seven months post first application in the first and second years, respectively. Treatment of *Trichoderma asperellum* in combination with composted plant residues through soil amendment, *Trichoderma* alone through foliar spraying, *Trichoderma* alone through stem infusion, and composted plant residues alone through soil amendment reduced this incidence. In the first year, a significant effect ($p \leq 0.05$) in VSD reduction compared to control was only observed three and seven months post treatment by *T. asperellum* plus composted plant residues. The other treatments did not have any significant impact to VSD reduction, except for *Trichoderma* infusion seven months post treatment (Figure 2). In the second year, significant

reductions of VSD ($p \leq 0.05$) compared to control were seen three months post treatment by *T. asperellum* plus composted plant residues and by composted plant residues alone. We also observed that VSD reduction compared to control five months post treatment by *T.asperellum* plus composted plant residues, *Trichoderma* spraying, and composted plant parts, and seven months post treatment by all treatments were significant. In this last month the VSD incidence in the groups treated by *T.asperellum* plus composted plant residues, *T. asperellum* spraying, *T.asperellum* infusion, and composted plant residue was 38.5%, 53.2%, 58.8%, and 50.6%, respectively, compared 69.2% in the control. The first treatment was significantly different ($p \leq 0.05$) from the other treatments except for the fourth treatment, there was no significant difference between second and fourth treatments, and these last two treatments were significantly different from the third treatment and control (Figure 2).

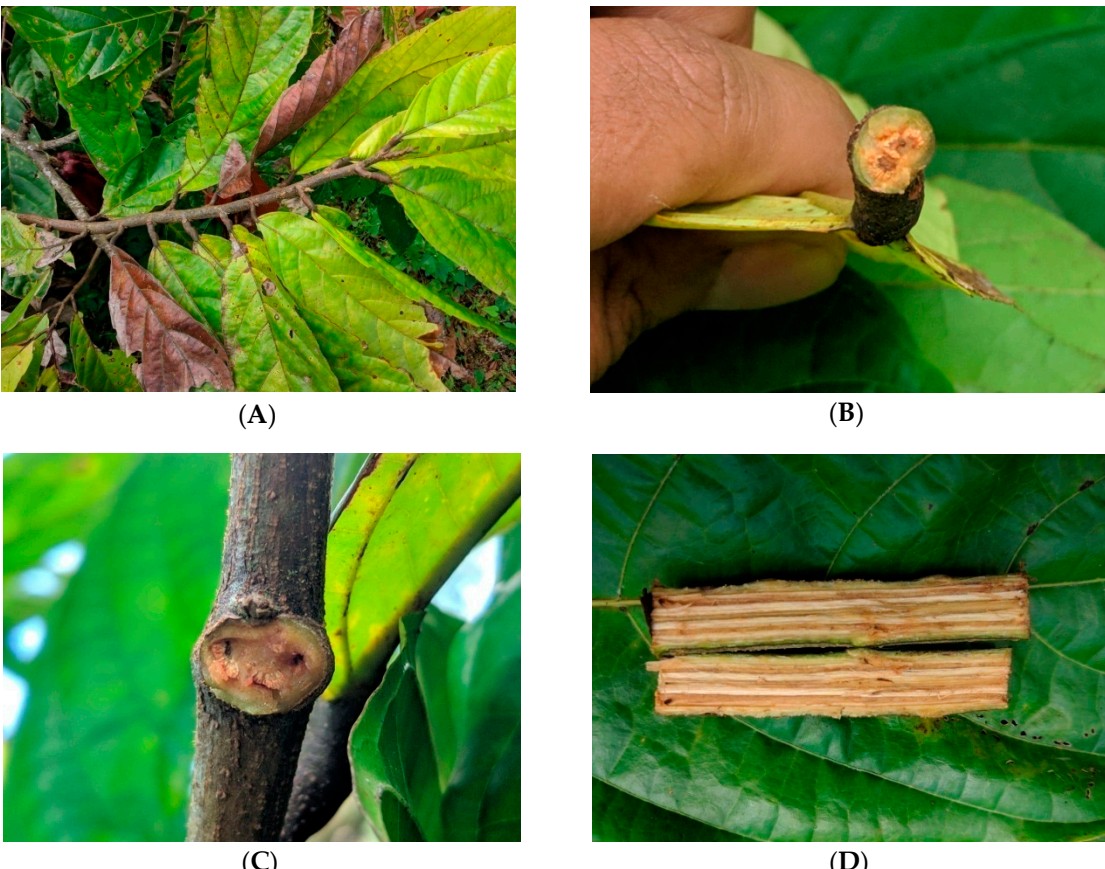

(A)      (B)

(C)      (D)

**Figure 1.** Symptoms of vascular streak dieback (VSD) disease. Full-leaf chlorosis and necrosison a branch (**A**) discoloration of vascular traces on petioles (**B**), on leaf scars (**C**), and in wood (**D**).

### 3.3. Re-Isolation of Trichoderma and Identification of Other Fungi in Root and Branch Tissues

*Trichoderma* could be detected in cacao treated with this fungus, except in those treated by stem infusion. In the trees not treated by the fungus, *Trichoderma* was either not detected or occurred at a level that was far below that observed in the treated trees. Therefore, we are sure that this isolated *Trichoderma* came from the applied fungus. Upon sampling at two months postfirst application of *Trichoderma* plus compost and leaf spraying, *Trichoderma* was detected in branch tissues with the colonization of 5% and 25% in the first year and 40% and 10%, respectively in the second year. In root tissues, colonization of *Trichoderma* with the same treatments was 20% and 0% in the first year and 5% and 10%, respectively, in the second year (Figure 3).

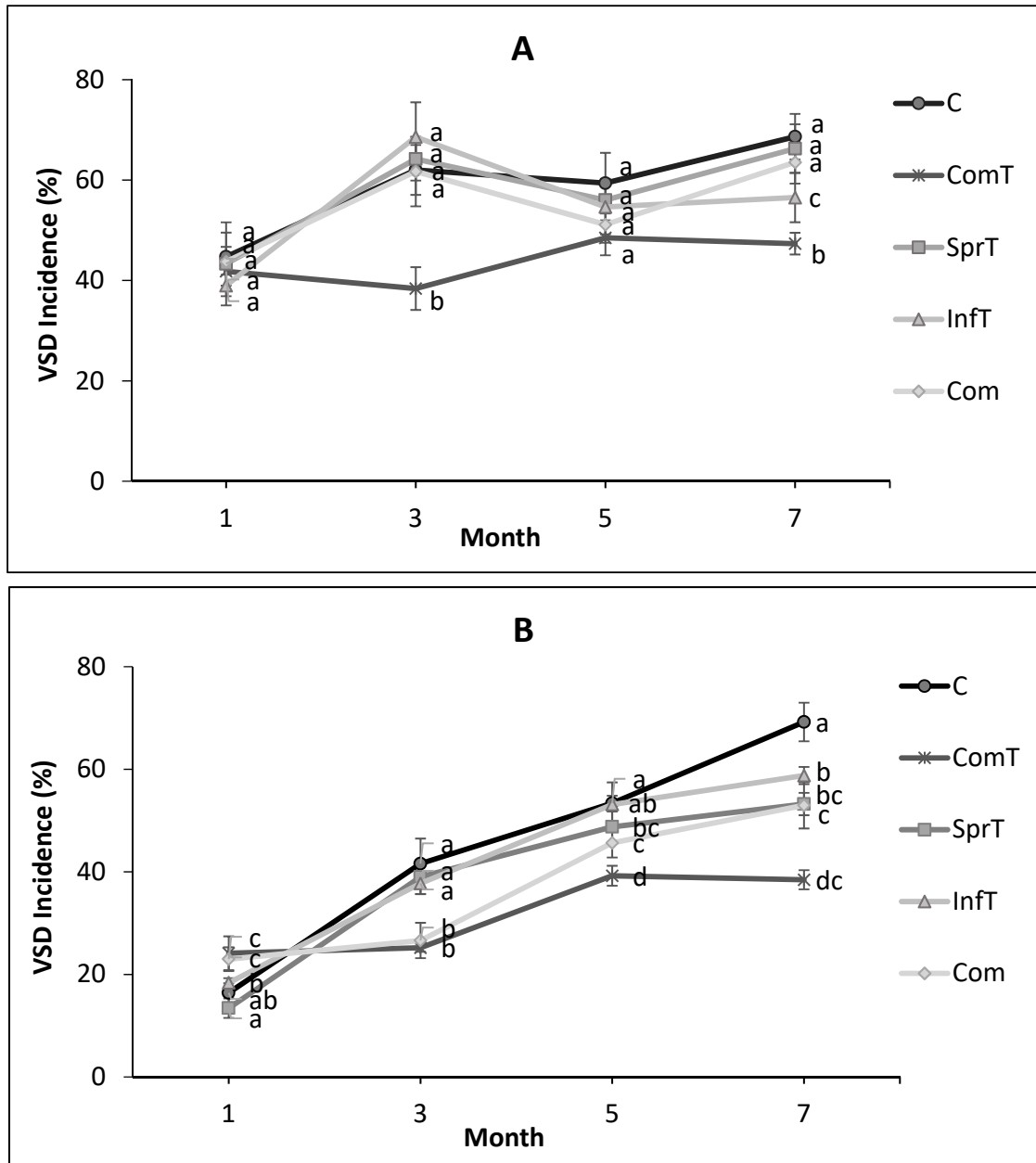

**Figure 2.** Dynamic of vascular streak dieback disease incidence on cacao heavy infected after treatment withcomposted plant residues and *Trichoderma asperellum* in the first year (**A**) and in the second year (**B**). C = control, ComT = compost plus *Trichoderma*, SprT = spraying of *Trichoderma*, InfT = infusion of *Trichoderma*, Com = compost. Means of incidence at the same time followed by the same letter are not significantly different according to LSDtest ($p \leq 0.05$).

Detection of co-occurring fungal endophytes in the treated and untreated trees indicated the presence of fungi such as *Lasiodiplodia* morphospecies 1, *Lasiodiplodia* morphospecies 2, *Lasiodiplodia* morphospecies 3, *Lasiodiplodia* morphospecies 4. *Lasiodiplodia* morphospecies 5, and *Fusarium decemcellulare*. In the first year, several fungi were found both in branch and root tissues of trees treated by *Trichoderma*, compost, *Trichoderma* plus compost, and control with the variation of colonization percentage. In the second year, we observed just one fungus (*Lasiodiplodia* morphospecies 3) occurring in branch and root tissues of control trees, while in trees treated *Trichoderma*, compost, and their combination, the presence of fungus was more diversified, comprising from three to five species, offering higher colonization compared to the control (Figure 3).

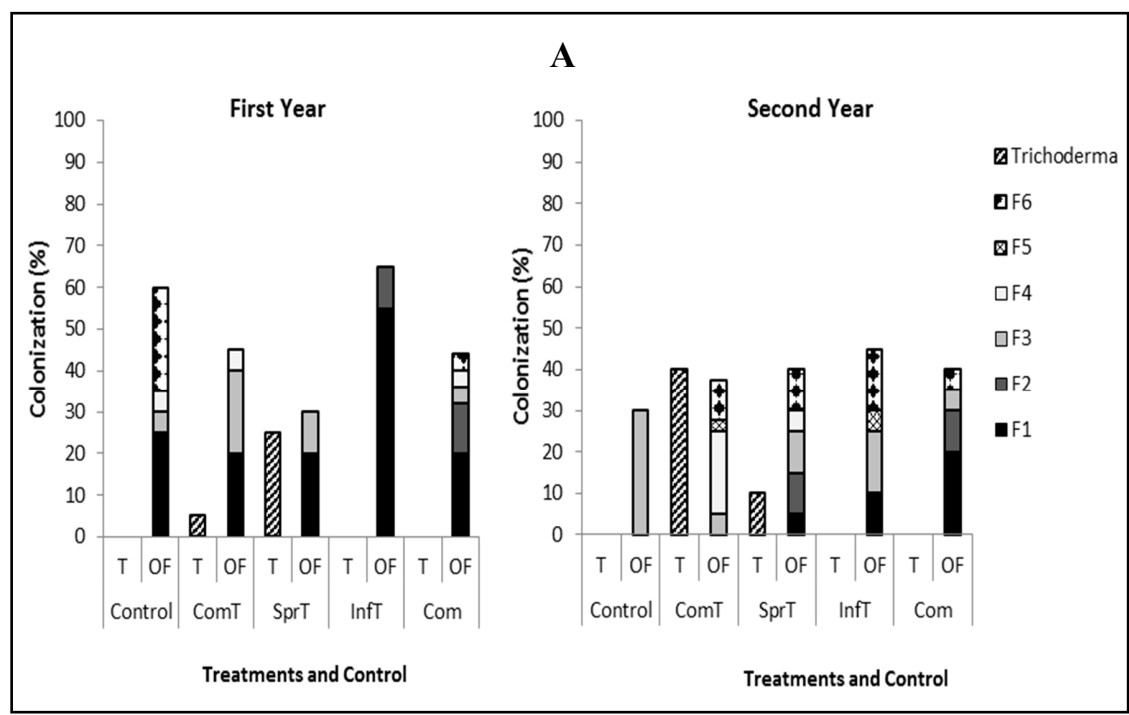

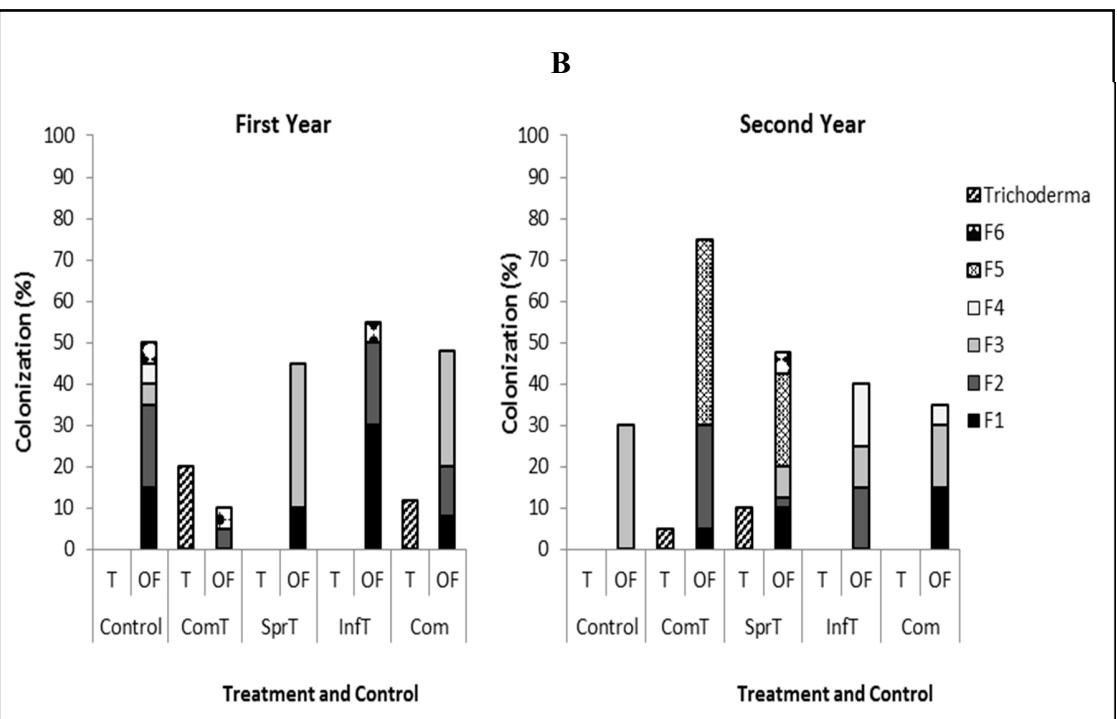

**Figure 3.** Colonization of *Trichoderma asperellum* and co-occurrence of endophytic fungi in branch tissues (**A**) and root tissues (**B**) in the first and second year. T = *Trichoderma,* OF = other fungi, ComT = compost plus *Trichoderma*, SprT = spraying of *Trichoderma*, InfT = infusion of *Trichoderma*, Com = compost, F1–F5 = *Lasiodiplodia* morphospecies 1–5, F6 = *Fusarium*.

## 4. Discussion

In addition to *Ceratobasidium theobromae*, we also identified some morphospecies of *Lasiodiplodia* and *Fusarium* in in branch tissues. These last two fungi have been known to cause dieback disease, and their symptoms include chlorosis and later necrosis of leaves, mostly those second or third from the

seedling stem tips. Abscission of these leaves occurs within two or three days of the first appearance of symptoms and dark discoloration on vascular tissues of wood is visible upon splitting the infected branches [11,13,14]. Vascular-streak dieback (VSD) shares the symptoms mentioned above, but the colonization of *Lasiodiplodia* and *Fusarium* both in root and branch tissues tended to increase in line with the reduction in incidence two years post-treatment (Figure 3).Therefore, symptoms presented in cacao trees of the Jalani clone was caused by VSD pathogen. Typical symptoms on cacao infected by this pathogen are leaf chlorosis, and some browning or necrosis of lamina tissue may occur with advanced chlorosis. In susceptible hosts, the tip of the branch dies, resulting in branch dieback. Another symptom different from those caused by other pathogens is brown-streaked vascular tissue, and the staining of three vascular traces in leaf scars [1,23]. The Jalani clone cacao showed full chlorosis in the leaf that developed rapidly to neighboring leaves, with an incidence of around 70% in seven months, as seen in untreated cacao. Then, visible brown-streaked vascular tissue and three black vascular traces in the leaf scars and the petioles were conspicuous (Figure 1), and the pathogen couldbe isolated from branches showing light symptoms. The high incidence observed in the field in two successive years indicated that the clone is susceptible to infection of *C. theobromae*.

Treatments with *T. asperellum* plus composted plant residues through soil amendment, *T. asperellum* alone through foliar spraying, *T. asperellum* alone through stem infusion, and composted plant residues alone through soil amendment in the first year indicated that the first treatment was the only one capable of reducing VSD incidence. However, its significant effect compared to the control fluctuated over seven months. In the second year, all treatments had a significant impact on the reduction of VSD incidence. This decrease occurred in the time span of three to seven months by the first treatment, five to seven months by the second and fourth treatments, and just in last month by the third treatment. Their efficacies were 44.4%, 23.1%, 15.1%, and 23.6%, respectively, around twenty months after the first treatment. In susceptible clones such as Jalani, rapid VSD development makes this disease more difficult to control and its reduction requires more than one time treatment. The capability of *T. asperellum* to reduce VSD incidence depends on the clone. Research done in Luwu Utara regency showed that the efficacy of *T. asperellum* in combination with composted plant residues on S1, MCC02, AP, THR, and RB clones was 60.1%, 55.2%, 49.9%, 83.3%, and 37,6%, respectively, six months post one-time treatment [24]. The reduction of incidence was due to a greater number of new healthy flush appearing in the tip compared to the increase of diseased leaves per branch. This successful field trial in controlling VSD disease followed what had been done in the laboratory [18] and related to the role of *Trichoderma* itself, compost itself, and a combination of both.

*T. asperellum* can deploy systemically after application through foliar spraying, stem infusion, and soil drenching, to almost all parts of the cacao seedling including roots, stem, and leaves. By application through foliar spraying and combination of *T. asperellum* and compost through soil amendment in this trial, we also identified the presence of this fungus in root and branch tissues. Therefore, the control mechanism hypothesis of *T. asperellum* against VSD pathogen is due to resistance induction through the interaction of this fungus with roots [25,26]. *T. asperellum* could also compete directly with the VSD pathogen in the infection site. A new finding indicates that the presence of *T. asperellum* in plant tissues increases the co-occurrence of endophytic fungi colonization such as *Fusarium*, *Lasiodiplodia*, and *Paecilomyces* [19]. In this trial, we also observed that colonization of *Lasiodiplodia* and *Fusarium* tended to increase both in roots and branches of trees treated by *Trichoderma* in the second year. This increase could have a role together with *T. asperellum* in inhibiting *C. theobromaee* directly or indirectly. Non published study on the relation between *T. asperellum* and anthracnose disease indicated that the decrease of this disease incidence was higher in cacao seedlings treated with *T.asperellum* plus *Fusarium decemcellulare* and plus *Lasiodiplodia pseudotheobromae* than with *T. asperellum* alone. *Fusarium* and *Lasiodiplodia* species are very diverse in cacao, and some are non-pathogenic, while others are pathogenic [8,12]. The pathogenic species express disease in the plant when in stress conditions [9,10,13].

A previous three-month study in the laboratory indicated that composted plant residues consisting of gliricidia leaf, rice straw, and billygoat weed did not have any significant effect on the reduction of VSD disease [18]. However, in this trial, the same composition of compost applied in the field through soil amendment could significantly reduce in the second year. This impact is apparently related to adequate dose and time of application for the compost to affect the metabolic processes in cacao. As *C. theobromae* infects the leaves and branches, and the compost is only in contact with the roots, the disease reduction phenomenon associated with compost should be systemic. Some researchers indicate that the inhibition of pathogen infecting aerial plant parts by compost is due to the induction of plant defense that shares similarities with both systemic acquired resistance and ABA-dependent/independent abiotic stress responses [27–29]. Microarray analyses of Arabidopsis plant infected by *Botrytis cinerea* fungus after treatment with the compost of olive marc and olive tree leaves revealed that 178 genes were differently expressed including those for biotic and abiotic stimulus, SA and ABA stimulus, systemic acquired resistance, and PR1 [27]. In addition, like the treatment with *T. asperellum*, treatment with compost also increased the co-occurring fungi in the second year, which could support the inhibition of the VSD pathogen.

The combination of *T. asperellum* and composted plant residues resulted in the highest decrease of VSD incidence, and this indicated a synergistic effect arising from a mixture of different mechanisms of the two treatments mentioned above. We do not know what supports this phenomenon, but *Trichoderma* is capable of growing and developing in composted plant residues [18], probably increasing the likelihood of penetrating and deploying in cacao tissues, and on the other hand, *Trichoderma* can degrade the compost making it available to root systems. Therefore, our results offer new insight to the application of the biological agent *T. asperellum* in combination with composted residues through soil amendment, because, since its discovery in the early 2000s in South Sulawesi [3], there has been no method capable of controlling the VSD disease effectively. An unpublished trial of flutriafol fungicide with the same clone used in this experiment resulted in a small impact on the reduction of VSD incidence.

## 5. Conclusions

We conclude that cacao of the Jalani clone in a field of the Bantaeng regency was infected by the VSD pathogen and was susceptible to the disease. *T.asperellum* and composted plant residues applied separately were capable of reducing the incidence of the disease. However, when the two were applied together through soil amendment, the combination was more effective in reducing the VSD incidence. Therefore, we suggest there are multiple modes of action in inhibiting the pathogen. A South Sulawesi government program for cacao including the complete replacement of non-productive and dead cacao is underway to accelerate the production of cacao in the region.The treatment could support this acceleration program through disease control and through the improvement of soil organic matter content and fertility. The content of soil organic matter in South Sulawesi is relatively low, around 1.45–1.83%, while soil nitrogen is 0.14% [30].

**Author Contributions:** Conceptualization, A.R.; methodology, A.R, M.T., and A.A.; formal analysis, A.R. and A.A.H.; investigation, A.R., A.A., and N.J.J.; resources, A.R.; data curation, A.A.H.; writing—original draft preparation, A.R.; writing—review and editing, A.R., M.T., and A.A.; project administration, N.J.J and A.A.; funding acquisition, A.R.

**Funding:** This research was funded by the Indonesian Agency for Agricultural Research and Development, Indonesian Ministry of Agriculture, grant number 54.15/HM.240/I.I/3/2016 and Hasanuddin University, grant number 3625/UN.4.21/LK.23/2017.

**Acknowledgments:** The authors are grateful to the Indonesian Agency for Agricultural Research and Development and Hasanuddin University who have supported this research through KKP3N and RUNAS grants, respectively.

**Conflicts of Interest:** The authors declare no conflicts of interest.

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
