# Peer review of "Dynamic of Vascular Streak Dieback Disease Incidence on Susceptible Cacao Treated with Composted Plant Residues and Trichoderma asperellum in Field"

_agronomy, doi:10.3390/agronomy9100650_

Round 1

Reviewer 1 Report

The authors have corrected and modified manuscript according to the reviewer's comments and provided additional preliminary information of about the raised queries. I can understand some of the raised concern cannot be fulfilled but I would be happy to see in coming future
The revised manuscript is substantially improved.

Author Response

Figures 3 and 4 which show four histograms have been put under one area Improvement of English is still in way

Reviewer 2 Report

Some points to further improve the manuscript: 1. the letters 'a, b, c' are not deleted from Figure 2 despite authors said that they have. It confuses the readers particularly when significance of these letters are not explained in the legend of the figure. 2. Many references are not written properly in the text: eg. Adu-Acheampong 2011 (line 58 and line 242); Barnett 1998 (line 153); Kiffer 1997 (line 153); Harmam or Harman 2011 (line 279); Sagara or Segarra (line 302); Vallad 2003 (line 303). Most of these references have more than one authors as per the reference list. These need attention. 3. del Castillo 2013 (line 290) is missing from the reference list. 4. The manuscript still suffers from English language at different sections. It can be improved.

Author Response

The letters a, b, c, indicate the significant different between treatment and these letter are explained in the legend of the Figure 2: Means of incidence at the same time, followed by the same letter are not significantly different, according to LSD-test (P≤05). References have been improved del Castillo et al. 2013 have been in Reference No 6. Could you mention which part that we could improve the English.

This manuscript is a resubmission of an earlier submission. The following is a list of the peer review reports and author responses from that submission.

Round 1

Reviewer 1 Report

VSD is an important disease for Cacao trees, which has significant economic impact. The MS entitled, "Dynamic of.............in field" can capture significant readers if it can be improved sufficiently. There are many areas of the MS which needs attention:

At various areas of the MS, sentences are not structured properly. In fact it is difficult to make the sense out of the sentence at various places. Even commas and full stops are not used correctly. The whole MS should be rewritten in a clear manner. It is more so in the results and discussion sections.

Abstract: The scientific names are not italicised

Introduction: Since different VSDs (Ceratobasidium, Fusarium and Lasiodiplodia sps) attack Cacao trees, it is desirable to explain Ceratobasidium relationship with other dieback diseases along with their significance.

Material and methods: The authors have used randomised block design but have not provided SD, SE and if they have used full randomisation. It is highly desirable to run the ANOVA on the disease ratings among the different times of observations and also for first and second year observations. 

M & M; Field assessment: Lines 138 on wards can be moved under sub-section of 'Analysis'

Results: There are figures 2, 3, 4, but in text figure 5 is also mentioned (line 222). In Figure 2, the letters a, b, c should be deleted from the plotted area. It confuses the reader as these letters are not shown or mentioned in the labels of the Figure or explained in its legend.

Figures 3 and 4 which show four histograms can be put under one area with proper explanations. These figures say 'T' and 'OF'. What does these mean? The figures should be self explanatory i.e. readers shouldn't read the text along side to understand these. 

Discussion: Needs to be rewritten in a clear and concise manner. It is difficult to follow in the present form. 

References: As many as 11 references which are mentioned in the MS are not listed under 'References'. To me, this is a glaring overlook on part of the authors. In the reference section, the numbering is not correct as well. 

Author Response

We would like to thanks very much for your review and correction. We fixed it and we add explanation in Figure 2 concerning significant difference between treatment.

Best Regards

Ade Rosmana

Reviewer 2 Report

It was pleasure reviewing the article entitled “Dynamic of vascular streak dieback disease incidence 2 on susceptible cacao treated with composted plant 3 residues and Trichoderma asperellum in field”. Rosmana et al. took the efforts in Trichoderma based bioformulation in controlling the Vascular streak dieback (VSD) disease caused by fungus Ceratobacidium theobromae. The manuscript rounded up clearly, but several experimental parts is missing to support their claims, which could further strengthen their field trials and observations. I would like to see following technical cavities in next revision before endorsing this manuscript for publications. Nevertheless, manuscript suffers from numerous typological error especially use of italics and punctuation marks, these amendments will significantly improve the quality of manuscript.

Comments

Authors have taken the account of morphological based analysis to identify isolated fungus Ceratobacidium theobromae, why they did not use species specific ITS based amplification, sequencing, biochemical characterization to confirm the fungal species. It should be noted that the authors focus heavily on field trail of Trichoderma asperellum strains, but why their antagonistic potential against Ceratobacidium theobromae has not been determined using the PDA plate assay (please refer the following manuscript (https://onlinelibrary.wiley.com/doi/full/10.1111/jph.12228). This would further strengthen their claims in field trail. In the re-isolation experiments and identification of occurrence of endophytes (Lasiodiplodia) only morphological attributes were considered for identification and characterization. Why not species level identification has not been performed using molecular-based detection system. Its very hard to convince that all five Lasiodiplodia are different. How authors can convince that Trichoderma asperellum doesn’t have any harmful effect on soil occurring beneficial microbes. It would be nice if authors will take one step forward to perform the isolation and quantification of microbes before and after treatment to know wider effect of bioformulation on soil microbes.      

Author Response

We would like to say thank you very much for your review and correction the manuscript number 551850. We have added what is lacking and  we add also the explanation concerning the significant different between treatment in Figure 2.

Best Regards,

Ade Rosmana